# Unveiling the Patterns and Drivers of Ecological Efficiency in Chinese Cities: A Comprehensive Study Using Super-Efficiency Slacks-Based Measure and Geographically Weighted Regression Approaches

**Jiantao Peng [1], Yihua Liu [1], Chong Xu [1,*] and Debao Chen [2]**

[1]  School of Geography and Remote Sensing, Guangzhou University, Guangzhou 510006, China; 2112101037@e.gzhu.edu.cn (J.P.); liuyihua@gzhu.edu.cn (Y.L.)
[2]  Department of Geography, University of Cincinnati, Cincinnati, OH 45221, USA; chen2do@mail.uc.edu
*   Correspondence: xchong@gzhu.edu.cn

**Abstract:** Urban ecological efficiency stands as a pivotal indicator that mirrors the level of sustainable development within cities. To unravel the sustainable development status of Chinese cities and illuminate the factors impacting the diverse developments among them, this study leveraged the super-efficiency SBM (slacks-based measure) model to assess the ecological efficiency of 284 prefectural-level and above cities across China in 2019, divulging their spatial distribution. Furthermore, a GWR (geographically weighted regression) model was also employed to scrutinize the factors influencing the ecological efficiency of these cities. Key findings include: (1) The mean ecological efficiency of Chinese cities in 2019 stood at 0.555, signaling moderate urban sustainability, with southern cities outperforming their northern counterparts. (2) A pronounced spatial clustering of ecological efficiency was evident, featuring positive spillover effects around high-efficiency cities and conversely, negative spillover effects around low-efficiency cities. (3) Economic development and population density positively influenced urban ecological efficiency, while urbanization levels exhibited a negative impact. The influences of industrial structure, technological level, and opening-up level varied, showcasing both positive and negative impacts contingent upon the spatial disposition of the cities. Hence, policymakers are advised to recognize the spatial nuances in the impacts of distinct factors on urban ecological efficiency and tailor measures accordingly to fortify urban sustainability.

**Keywords:** sustainable development; ecological efficiency; spatial autocorrelation; GWR model; influencing factors

## 1. Introduction

Over the past forty years, China has undergone a remarkable surge in socioeconomic transformation, fueled by comprehensive reforms and its policy of opening up. This period witnessed rapid urbanization and extensive urban expansion. However, these strides have also brought forth substantial challenges related to resources and the environment, manifesting in issues like resource depletion, ecological degradation, and pollution. These challenges are direct outcomes of the prevailing mode of development and pose significant threats to China's pursuit of sustainable development.

Amidst the array of challenges and pressures confronting China, the government has undertaken substantial initiatives in recent years to address sustainability concerns. At the macro level, sustainability has been entrenched as a foundational strategy since 1997, with recent emphasis placed on visionary objectives such as carbon peaking and carbon neutrality. Concurrently, the central government has underscored the imperative of prioritizing sustainable development policies within regional development frameworks. For instance, the Guangdong–Hong Kong–Macao Greater Bay Area has proposed the implementation of a stringent ecological protection regime within its developmental blueprint,

with a concerted focus on rectifying historical environmental legacies. Similarly, Northeast China has enacted policies mandating the cessation of natural forest logging, diverting attention towards leveraging natural and ecological resources for tourism purposes, thereby facilitating the transition towards sustainable development paradigms.

Furthermore, the government has embarked on initiatives aimed at establishing sustainable development demonstration zones across the nation. For example, Shenzhen City has emerged as an exemplar of sustainable development practices within the context of mega-cities, while Taiyuan City serves as a model for the transformation and rejuvenation of resource-based urban centers. Guilin City, on the other hand, has been designated as a demonstration zone for the sustainable utilization of landscape resources. The knowledge and insights gleaned from the experiences of these model zones in advancing sustainable development have been disseminated not only across the nation but also globally, underscoring China's commitment to fostering sustainable development at both domestic and international levels.

Moreover, the objective and scientific measurement of the current state of sustainable development of a region or city and what factors are affected by it is an important foundation and basis for us to continue to reform and upgrade. Ecological efficiency, in recent years, has been an effective tool used to measure the level of sustainable development. The study of ecological efficiency is conducive to improving our understanding of sustainable development. In other words, scientifically and effectively evaluating the level of ecological efficiency in China's cities, discovering the differences in the level of ecological efficiency, analyzing in depth the factors affecting the level of regional ecological efficiency, and exploring ways to improve regional ecological efficiency are of great significance to China's efforts to positively transform its mode of economic development, harmonize the inherent contradictions of the composite system of economy–resources–environment, and realize sustainable development.

The inception of the concept of ecological efficiency dates back to 1989 when Schaltegger and Sturm introduced it as the ratio of economic growth to environmental impact [1]. This concept, with its simplicity and significance, prompted numerous organizations and scholars to delve further, leading to multiple definitions and expansions of the concept [2–4]. Among these, the definition established by the WBCSD (World Business Council for Sustainable Development) is widely accepted by the public. It defines ecological efficiency as the process of gradually reducing ecological impacts and resource intensities throughout the procedure, aiming to align with the Earth's estimated carrying capacity while offering products and services that meet human needs and enhance the quality of life [5].

While academic consensus on the precise definition of ecological efficiency remains elusive, there exists a shared core idea across varied definitions: the optimization of economic returns with minimal resource and environmental consumption, driving sustainable economic development. The concept of ecological efficiency, thus, serves as an equilibrium point between economic development and resource-environmental concerns, emerging as a crucial indicator for gauging sustainable regional development. Studies revolving around ecological efficiency yield invaluable insights into the sustainable evolution of human society.

Over the past three decades, academic research on ecological efficiency has expanded rapidly and achieved fruitful results. Other countries and China have significantly different focuses on ecological efficiency research. In foreign countries, scholars focus more on the research object of ecological efficiency on enterprise operation [6,7] and industry development [8,9]. In contrast, scholars in China have focused more on the three major industries [10–15] as well as regional research on ecological efficiency, especially regional research on ecological efficiency. This is determined by the actual situation and real needs of China's social and economic development over the years.

In the realm of research on regional ecological efficiency in China, scholars have directed their investigations towards various geographic scales, including provinces [16,17],

watersheds [18,19], and urban agglomerations or economic zones comprised of multiple cities [20,21]. Research endeavors have predominantly focused on three principal facets.

Firstly, there is a concerted effort towards the measurement of ecological efficiency, which serves as a foundational component underpinning subsequent inquiries. A plethora of methodologies have been employed for this purpose, ranging from the ecological footprint method utilized by Shi [22] to the energy value analysis method employed by Pan [23], as well as the construction of ecological efficiency evaluation index systems by Qiu [24] and Li [25]. Furthermore, modeling approaches such as the SFA (Stochastic Frontier Analysis) model and DEA (Data Envelopment Analysis) models, employed by Yang [26] and Ren [27], respectively, have gained prominence over time. Secondly, attention has been devoted to scrutinizing the spatiotemporal dynamics and patterns of ecological efficiency. Mathematical models including the Markov chain [28] and spatial autocorrelation [29] have been employed to unveil the temporal evolution of regional ecological efficiency, often aided by visualization tools such as GIS. Lastly, scholars have endeavored to elucidate the influencing factors shaping ecological efficiency. By selecting specific determinants in line with the conceptual framework of ecological efficiency, researchers have explored the correlation between these factors and ecological efficiency through regression modeling techniques. Commonly employed regression models include the Tobit model [26,30], Geodetector [31], and spatial measurement model [32,33]. Notably, numerous representative studies have underscored the significance of factors such as economic development, industrial structure, technological advancement, and level of openness in influencing urban ecological efficiency. Nevertheless, divergent results across specific research domains and temporal contexts have been observed, owing to disparities in indicator selection and assessment methodologies utilized across studies.

Extensive literature review underscores the significance of scientifically rigorous assessment of urban ecological efficiency and examination of its spatial and temporal distribution, along with its influencing factors, to drive sustainable urban development forward. While past studies have contributed substantially to elucidating the interplay between resource and environmental inputs and economic outputs within the framework of social development, notable challenges remain.

Firstly, existing nationwide studies have predominantly utilized the province as the primary unit for ecological efficiency assessment, potentially neglecting regional variations in urban development. Therefore, there is a pressing need to prioritize city-level ecological efficiency studies within China. Secondly, past research has frequently employed global models like Geodetector and Tobit models to examine the factors influencing ecological efficiency. However, these models assume that influencing factors are independent of the geographic location of the research subject. In reality, regression parameters often exhibit substantial variations across different geographic locations, making these approaches inadequate for capturing regional disparities in ecological efficiency.

To furnish a more comprehensive understanding of sustainable development in Chinese cities, this study employs the super-efficiency SBM (Slacks-Based Measure) model to gauge the ecological efficiency of these cities. Furthermore, the study introduces the GWR (Geographically Weighted Regression) model to scrutinize the factors influencing urban ecological efficiency and its heterogeneity. The super-efficiency SBM model, grounded in establishing input–output indicator systems, stands as the predominant method for assessing ecological efficiency. Conversely, the GWR model, a variant of the local regression model, integrates geographic information into regression parameters to elucidate spatial disparities among influencing factors across different geographical locations. Despite the widespread use of the GWR model, its incorporation into studies on factors influencing ecological efficiency remains limited. Therefore, this study aims to bridge this gap by employing the GWR model to investigate influencing factors, with the objective of generating insights beyond those gleaned from prior research endeavors.

Through these assessments, the study aims to provide scientific references that can facilitate the formulation of diverse regional strategies for promoting sustainable socio-economic development in Chinese cities.

## 2. Methods and Data Sources

### 2.1. Methodology

Ecological efficiency stands as a pivotal measure of sustainable development, encapsulating harmonious and integrated performance across resource utilization, environmental impact, and economic vitality. This study, rooted in an extensive literature review, adopted a unified definition of ecological efficiency, highlighting the holistic interplay among these three fundamental aspects. To measure ecological efficiency, this study employed the super-efficiency SBM model, which is a comprehensive evaluation method enabling a comprehensive depiction of sustainable development. Furthermore, considering the multiple factors influencing ecological efficiency across diverse regions, this study integrated the GWR model for further deep assessment. Unlike global spatial models, GWR accounts for spatial variations in the impact of each factor, offering a more precise identification of ecological efficiency determinants. This study embarked on exploring and delineating these intricate interconnections based on the conceptual framework in Figure 1.

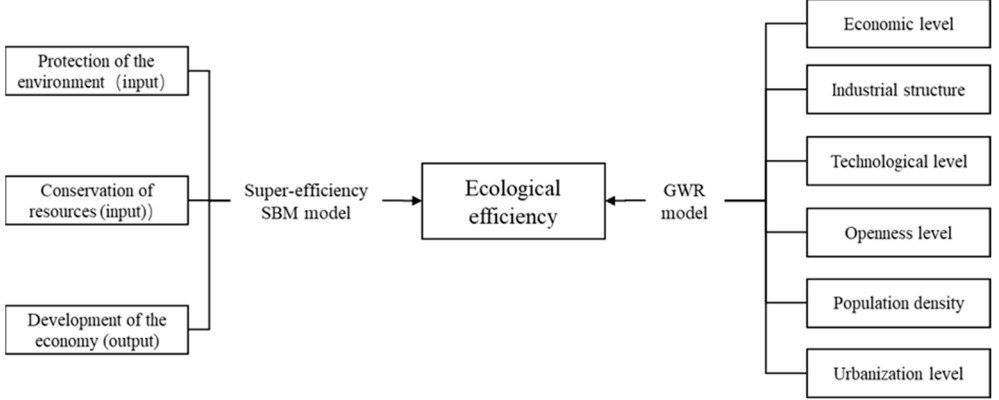

**Figure 1.** Theoretical framework for ecological efficiency assessment.

### 2.1.1. Super-Efficiency SBM Model

Our literature review reveals three primary methods for measuring ecological efficiency: the single ratio method, the indicator system method, and the modeling method. The single ratio method, which relies on just two indicators, tends to provide a limited perspective on ecological efficiency. The indicator system method, although more comprehensive, often struggles with subjectivity during analysis and in the weighting of calculations. Over time, the modeling method has emerged as the predominant approach for its comprehensive and objective evaluation of ecological efficiency. Among these, the DEA model stands out as the most extensively applied technique.

DEA is a non-parametric method that evaluates the relative efficiency of entities with multiple inputs and outputs, introduced by Charnes et al. in 1978 [34]. Its main advantage lies in its objective analysis, eliminating the need for predetermined weights for inputs and outputs. However, traditional DEA models have limitations, notably their inability to account for slack variables in ineffective DMUs (decision-making units). To overcome this, Tone introduced the SBM model in 2001 [35], a non-radial and non-angular approach that incorporates slack variables. While the SBM model successfully addresses the issue of slack variables, it introduces a challenge by often yielding uniform efficiency scores across decision units, which complicates rankings and further analysis due to all efficiency values being equal to 1. To resolve this, Tone developed the super-efficiency SBM model in 2002 [36], which provides a nuanced evaluation of efficiency among different units by

solving the issue of uniform efficiency scores. The formulation of the super-efficiency SBM model is as follows:

$$minp = \frac{1 + \frac{1}{m} \sum\limits_{i=1}^{m} s_i^- / x_{ik}}{1 - \frac{1}{s} \sum\limits_{r=1}^{s} s_r^+ / y_{rk}} \tag{1}$$

$$s.t. \sum\limits_{j=1,j \neq k}^{n} x_{ij} \lambda_j - s_i^- \leq x_{ik} \ (i = 1, 2, \ldots, m)$$

$$\sum\limits_{j=1,j \neq k}^{n} y_{rj} \lambda_j + s_r^+ \geq y_{rk} \ (r = 1, 2, \ldots, m)$$

$$\lambda_j \geq 0, j = 1, 2, \ldots, n \ (j \neq k), s - i \geq 0, s + r \geq 0$$

where *minp* represents the relative efficiency; $x$ and $y$ denote the input and output variables, respectively. $m$ and $s$ represent the number of input and output indicators. $S_i^-$ and $S_r^+$ represents the slack variables for input and output, respectively. $\lambda_j$ is the weight vector. The evaluated decision unit is considered relatively efficient if the relative efficiency is greater than 1.0, and relatively inefficient if the relative efficiency value is lower than 1.0, indicating that input and output adjustments are required.

2.1.2. Spatial Autocorrelation Analysis

The "First Law of Geography" posits that spatial proximity fosters stronger correlations among entities compared to those farther apart [37]. Spatial autocorrelation analysis, as a method to assess the degree of spatial clustering for a specific attribute within spatial units, encompasses two types: global and local measurements. Global spatial autocorrelation depicts the overall spatial correlation within an entire region, while local spatial autocorrelation gauges spatial relationships among sub-regions. In this study, we utilized both global and local spatial autocorrelation to investigate the spatial clustering of urban ecological efficiency [38]. The global Moran's Index evaluates the overall spatial clustering across the entire spatial sequence, while the local Moran's Index assesses clustering concerning neighboring areas within specific sub-regions. Their respective formulas are as follows:

$$I = [\sum\limits_{i=1,j=1}^{n} W_{ij}(x_j - x)(x_j - x)] / S^2 \sum\limits_{i=1,j=1}^{n} W_{ij} \tag{2}$$

$$I_i = [\sum\limits_{i=1,j=1}^{n} W_{ij}(x_j - x_i)] / S^2 \tag{3}$$

where $I$ represents the global Moran's Index; $S^2 = \frac{1}{n} \sum\limits_{i=1}^{n} (x_i - \bar{x})^2$ denotes the sample variance. $n$ is the number of study units (number of cities in this study). $x_i$ and $x_j$ represent the ecological efficiencies of cities $i$ and $j$, respectively, while $\bar{x}$ represents the mean ecological efficiency of all cities. $W_{ij}$ represents the spatial weight matrix. The global Moran's Index $I$ ranges between $-1$ and 1. A positive value indicates positive spatial autocorrelation, while a negative value indicates negative spatial autocorrelation. $I_i$ represents the value of the local Moran's Index. By plotting a local Moran scatterplot, the spatial clustering of ecological efficiency for each city can be visually displayed. The local spatial correlation of regions can be visualized by plotting LISA clustering maps in four cases: "High–High (H-H)" clustering, which means that cells with high observations are also surrounded by cells with high observations; "Low-High (L–H)" clustering, which indicates that cells with low observations are surrounded by cells with high observations; "low-low (L–L)" clustering, which indicates that cells with low observations are surrounded by cells with low observations; The "High-Low (H–L)" clustering indicates that cells with high observations are surrounded by cells with low observations.

### 2.1.3. GWR Model

It is evident that urban ecological efficiency possesses geospatial attributes, necessitating a regression model that can adeptly handle data with spatial characteristics. Brunsdon et al. introduced the GWR model, employing local smoothing techniques to account for these attributes [39]. Unlike traditional models that rely on global averaging, GWR acknowledges the variability in local changes associated with regional locations, incorporating geographical information of sample points into parameter estimations. Consequently, GWR proves highly effective for analyzing the determinants of urban ecological efficiency. The formula for the GWR model is presented as follows:

$$y_i = \beta_0(u_i, v_i) + \sum_{k=1}^{p} \beta_k(u_i, v_i)x_{ik} + \varepsilon_i \tag{4}$$

In the equation: $y_i$ represents the dependent variable's explanatory value for city $i$; $(u_i, v_i)$ represents the geographic coordinates of city $i$; $x_{ik}$ represents the explanatory values of the independent variables for city $i$; $\beta_k(u_i, v_i)$ represents the regression parameters at the centroid $(u_i, v_i)$ of research unit $i$, which is a function of geographical location; and $\varepsilon_i$ represents the random error term.

To avoid the bias of the estimation results caused by the interaction between the indicators, it is necessary to carry out the covariance test of the above indicators before carrying out the geographically weighted regression, and if there is a strong covariance of the indicators, it is necessary to exclude them. At the same time, the data need to be standardized before regression to maintain the smoothness of the data.

### 2.2. Ecological Efficiency Measurement Indicators and Data Sources

### 2.2.1. Construction of the Ecological Efficiency Measurement Indicator System

Developing a scientific and comprehensive indicator system for measuring urban ecological efficiency is crucial for research, necessitating an analysis rooted in the concept of ecological efficiency. Urban ecological efficiency can be succinctly defined as the ratio of a city's economic output to its resource and environmental consumption, encompassing three key dimensions: economy, resources, and environment. Drawing from the analysis of seminal literature [26,27,40], this study similarly adopts these three dimensions in designing the ecological efficiency measurement indicator system: economic, resource, and environmental pollution categories. The economic category serves as the output indicator, resource consumption as the input indicator, and environmental pollution, though an outcome of economic activities, is considered an additional input factor for its impact on the environment.

(1) Economic indicators primarily capture the value of products and services generated by the economic system, represented in this study by each city's GDP (Gross Domestic Product). (2) Resource-based indicators encompass a broad range of material, financial, and human elements available within a country or region, categorized into natural and social resources. This study focuses on three natural resources—energy, water, and land—due to their significant relevance to human economic activities. Energy consumption is indicated by the city's total electricity consumption; water consumption by the social water consumption index; and land resources by the area of urban construction land. Social resources are represented by labor and capital, measured by the number of employees and fixed asset investment, respectively. (3) Environmental impact indicators are quantified by the societal emissions of wastewater, exhaust gas, and solid waste—the "three wastes". Given the lack of comprehensive data publication in China, this study follows the approach of most Chinese researchers, selecting sewage discharge, industrial sulfur dioxide emissions, and garbage generation as markers for environmental pollution indicators (Table 1).

**Table 1.** The indicator descriptions for the ecological efficiency evaluation.

| Indicator | Composition of Indicator | Concrete Content | Unit |
|---|---|---|---|
| Input | Resource consumption | Total electricity consumption | KWh |
| | | Total water consumption | $10^4$ tons |
| | | Construction land area | $km^2$ |
| | Labor force consumption | Number of employed personnel | $10^4$ People |
| | Capital consumption | Fixed asset investment | $10^4$ Yuan |
| | Environmental consumption | Sewage discharge | Ton |
| | | Industrial sulfur dioxide emissions | Ton |
| | | Garbage generation | Ton |
| Output | City economic output | GDP of each city | $10^8$ Yuan |

### 2.2.2. Factors Affecting Ecological Efficiency

The city represents a holistic and intricate system, encapsulating elements of ecology, economy, and environment. The determinants of urban ecological efficiency are multifaceted, prompting this study to undertake indicator selection from two main perspectives. Firstly, leveraging insights from prior literature, we conducted a comprehensive review of recent scholarly work on urban ecological efficiency, synthesizing and categorizing identified influencing factors. Secondly, informed by the essence of ecological efficiency and the principles of sustainable development, we refined our indicator selection process. An examination of key literature [26,32,40,41] reveals a consensus among researchers on several core factors affecting urban ecological efficiency, including economic level, industrial structure, level of openness, technological advancement, and urbanization rate. Additional considerations by some scholars include urban compactness and marketization degree. Therefore, grounded in the aforementioned analysis and the principle of data availability, this study opts to explore six dimensions—economic level, industrial structure, technological advancement, level of openness, population density, and urbanization rate—as the pivotal influencers of urban ecological efficiency. Table 2 presents the specific indices used to characterize each dimension.

**Table 2.** Factors affecting urban ecological efficiency.

| Factor | Measurement Methods | Unit |
|---|---|---|
| Economic level | GDP per capita | Yuan |
| Industrial structure | GDP of secondary industry/GDP | % |
| Technological level | Science and Technology expenditures/Financial expenditure | % |
| Opening-up level | Foreign direct investment/GDP | % |
| Population density | Population/Urban built-up area | People/$km^2$ |
| Urbanization level | Urban population/Total population | % |

### 2.2.3. Data Sources

The research object of this study is Chinese cities at the prefecture level and above, but the following types of regions were excluded from the study due to serious data missing: (1) Hong Kong, Macau, and Taiwan, the three regions outside mainland China. (2) Seven prefectures, such as Da Hinggan Ling Prefecture and Kashgar Prefecture; three leagues, including Xilingol League, Alxa League, and Hinggan League; and thirty Autonomous Prefectures, such as Yanbian Korean Autonomous Prefecture and Enshi Tujia and Miao Autonomous Prefecture. (3) Thirteen prefecture-level cities such as Xiantao and Tianmen. These regions are mainly located in the western part of China such as Xinjiang, Tibet, Qinghai, Sichuan, Yunnan, and Guizhou. Finally, 280 prefecture-level cities and four municipalities totaling 284 cities were selected for this study, as shown in Figure 2.

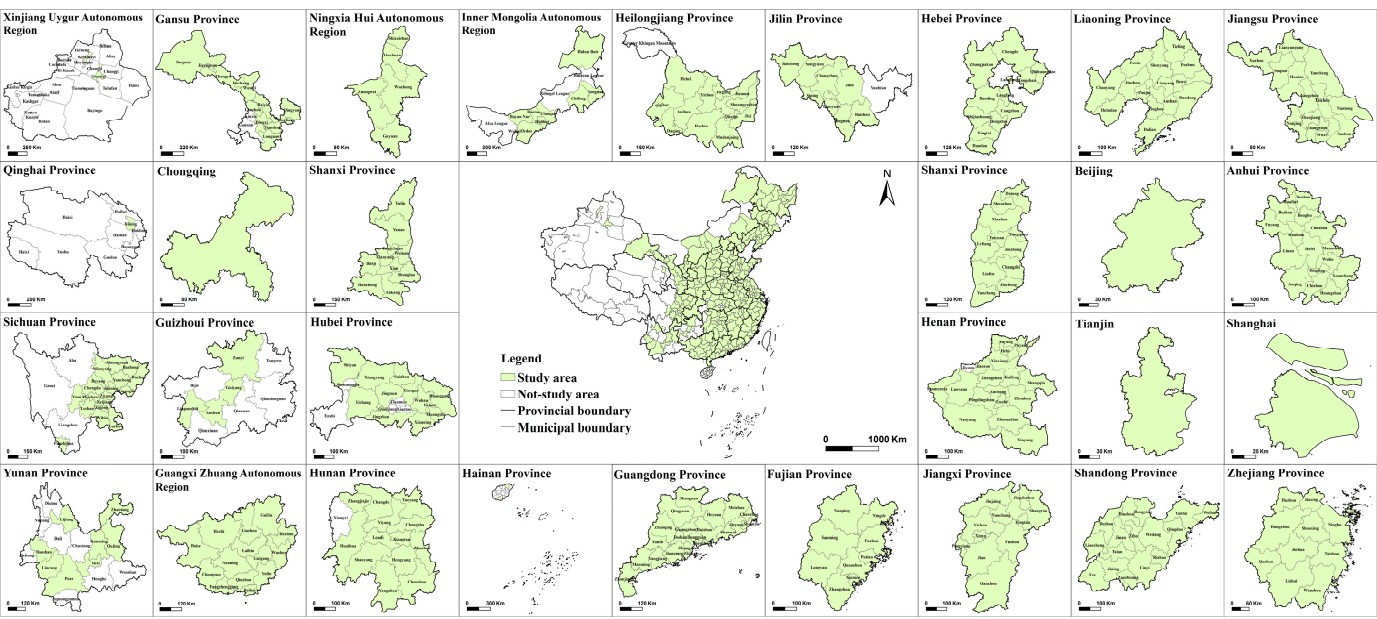

**Figure 2.** Study area.

This study aims to study the ecological efficiency of Chinese cities in recent years, and the data used are all social and economic data. However, there has been a worldwide outbreak of the COVID-19 Pandemic since 2020, which has had a great impact on the economy and society, and the related data have also changed abnormally. For the sake of scientificity and accuracy, this study adopts the data of 2019, and the data mentioned above are all obtained from the 2020 China Urban Statistical Yearbook (which reflects the economic data of 2019), the 2019 China Urban Construction Statistical Yearbook and the statistical yearbooks of each province, and the sources of specific indicators are detailed in Table 3. The few outliers in the yearbooks are adjusted through the data of the official regional website or supplemented by the use of the interpolation method. Finally, descriptive statistics are provided for the data in this study, as shown in Table 4.

**Table 3.** List of sources of data.

| Data Sources | Year of Data | Name of Data | Units |
|---|---|---|---|
| China Urban Statistical Yearbook | 2020 | GDP of each city | $10^8$ Yuan |
| | | GDP per capita | Yuan |
| | | Fixed asset investment | $10^4$ Yuan |
| | | Number of employed personnel | $10^4$ People |
| | | Industrial sulfur dioxide emissions | Ton |
| | | Sewage discharge | Ton |
| | | Garbage generation | Ton |
| | | Financial expenditure | $10^8$ Yuan |
| | | Science and Technology expenditures | $10^8$ Yuan |
| | | Foreign direct investment | $10^4$ Dollar |
| China Urban Construction Statistical Yearbook | 2019 | Construction land area | km$^2$ |
| | | Urban built-up area | km$^2$ |
| | | Total water consumption | $10^4$ tons |
| Provincial statistical yearbooks | 2020 | Total electricity consumption | KWh |
| | | Industrial structure | % |
| | | Population | $10^4$ People |
| | | Urbanization level | % |

**Table 4.** Data descriptive statistics.

| Name of Data | Units | Maximum | Minimum | Mean Value |
|---|---|---|---|---|
| GDP of each city | $10^8$ Yuan | 38,156.00 | 231.00 | 3326.07 |
| GDP per capita | Yuan | 203,489.00 | 14,746.00 | 63,542.69 |
| Fixed asset investment | $10^4$ Yuan | 19,724.00 | 72.00 | 2325.43 |
| Number of employed personnel | $10^4$ People | 791.30 | 5.86 | 59.74 |
| Industrial sulfur dioxide emissions | Ton | 115,089.00 | 75.00 | 11,587.27 |
| Sewage discharge | Ton | 96,501.00 | 48.00 | 4405.27 |
| Garbage generation | Ton | 213,693.00 | 79.00 | 14,323.64 |
| Financial expenditure | $10^8$ Yuan | 8179.28 | 31.86 | 574.82 |
| Science and Technology expenditures | $10^8$ Yuan | 548.42 | 0.18 | 18.39 |
| Foreign direct investment | $10^4$ Dollar | 190.47 | 0.00 | 9.53 |
| Construction land area | $km^2$ | 1495.00 | 13.00 | 197.00 |
| Urban built-up area | $km^2$ | 1515.41 | 13.80 | 202.68 |
| Total water consumption | $10^4$ tons | 297,923.20 | 763.00 | 21,403.70 |
| Total electricity consumption | KWh | 1568.58 | 24.96 | 229.99 |
| Industrial structure | % | 67.04 | 10.68 | 39.46 |
| Population | $10^4$ People | 3113.00 | 30.49 | 460.37 |
| Urbanization level | % | 99.52 | 34.67 | 60.40 |

## 3. Results

### 3.1. Urban Ecological Efficiency Assessment

Using the data from 284 Chinese cities in 2019, the ecological efficiency of these cities was measured using the super-efficiency SBM model of MAX-DEA 9.1 software. Regarding the efficiency measurement division standard of existing studies [28], they were divided into low efficiency, medium efficiency, relatively high efficiency, and high efficiency with critical values of 0.50, 0.75, and 1. The results show that the average value of ecological efficiency of Chinese cities is 0.555, which is in the state of medium efficiency, and there are only 33 cities that reach high efficiency, with the overall low ecological efficiency of cities and poor coordination between economic development and nature and ecology. To explore the ecological efficiency status of Chinese cities from different perspectives, this study will analyze three perspectives: the overall situation, the four major economic zones, and cities at multiple scales.

(1) Overall situation. The measurement results were visualized using ArcGIS 10.8 software, as shown in Figure 3a. From the overall spatial distribution, the ecological efficiency of cities in southern China is significantly better than that of northern cities. The number of cities with high ecological efficiency is small, but most of them have obvious clustering characteristics. For example, the southern part of Henan Province (Figure 3(b1)), the eastern and southern parts of Jiangsu Province (Figure 3(b2)), Fujian Province (Figure 3(b3)), and the core cities of the Pearl River Delta (Figure 3(b4)) are the main areas of high ecological efficiency distribution. Relying on the advantages of location and policies, these cities have achieved high economic outputs. With advanced management measures, they have gained a first-mover advantage in sustainable development and have formed a high ecological efficiency agglomeration. High ecological efficiency agglomerations have also been formed in several cities neighboring Inner Mongolia, Shanxi and Gansu provinces (Figure 3(b5)), which have achieved high ecological efficiency by virtue of their lower resource and environmental consumption, even though their economic output is not high.

Cities with low ecological efficiency show more obvious agglomeration characteristics in spatial distribution. The ecological efficiency of cities in several provinces in a row, such as Jilin, Heilongjiang, Liaoning, Hebei, Shanxi, Shandong, etc. (Figure 3(c1)), is almost at the low-efficiency level, and only Beijing and Yangquan are at the high-efficiency status in these regions. Yantai and Qingdao in Shandong are at a relatively high efficiency status. These cities located in northeastern China have been relying on heavy industries for a long time for their economic development, with serious resource consumption and

environmental pollution. With a gradual loss of population in recent years, the vitality of urban development and the rate of economic growth has declined, resulting in the formation of low-efficiency agglomerations. Cities in the provinces of Inner Mongolia, Gansu, and Ningxia Province (Figure 3(c2)) are also in a state of low ecological efficiency; in addition, there are also many cities with low ecological efficiency concentrated in several provinces, such as Jiangxi, Guangdong, Guangxi, and Yunnan Province (Figure 3(c3)). Mainly due to the low level of economic development, the cities located in the northwest and southwest of the study area maintain current low-efficiency levels.

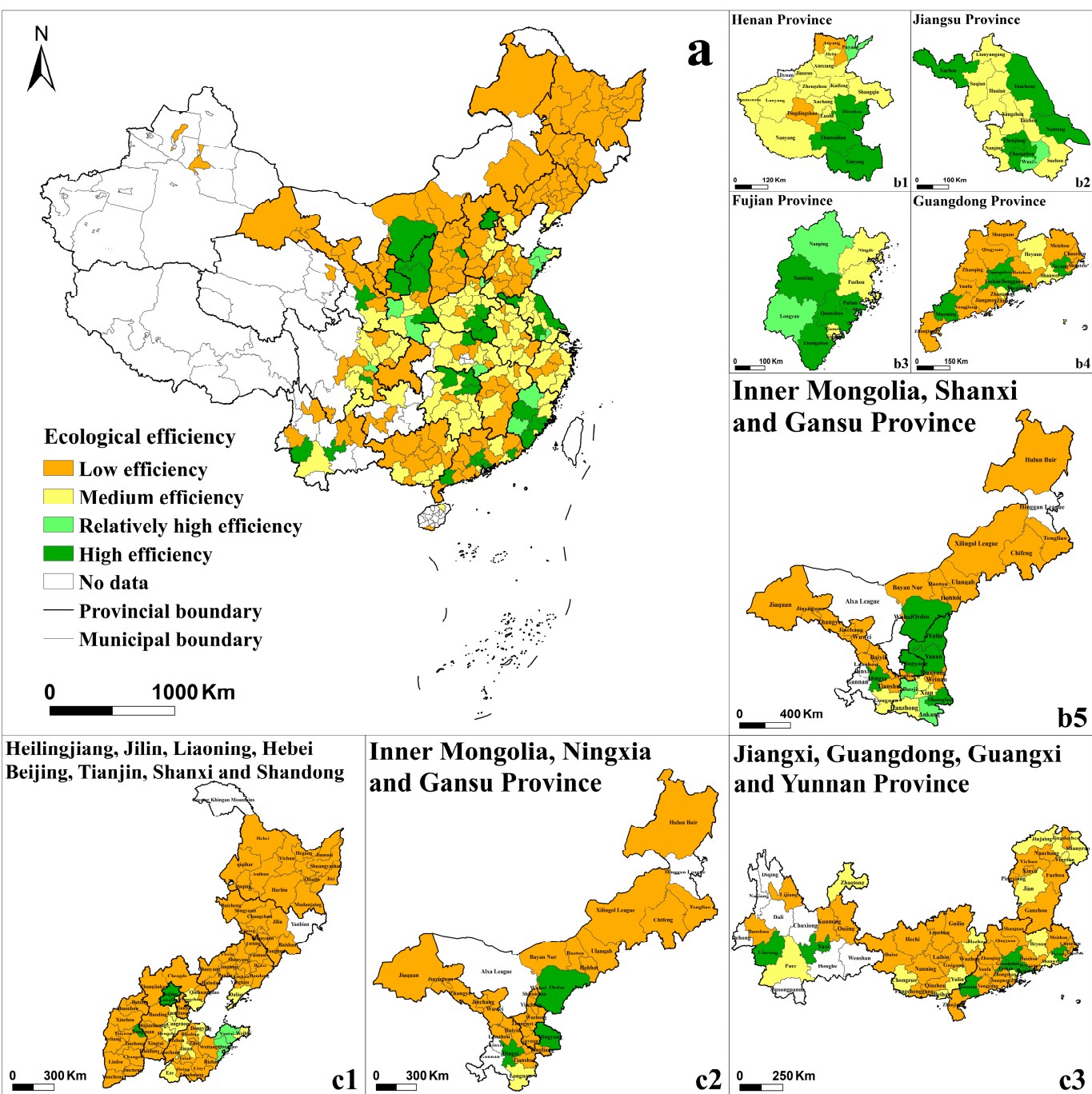

**Figure 3.** Spatial configuration of ecological efficiency in Chinese cities. (**a**). Overall ecological efficiency of Chinese cities. (**b1**–**b5**). Distribution areas of high ecological efficiency. (**c1**–**c3**). Distribution areas of low ecological efficiency.

China's eastern coastal provinces are more economically developed, but most of these cities are not in a state of high efficiency. In addition, like the inner distribution of Guangdong Province, the cities located in the center (Guangzhou, Foshan, Shenzhen, etc.) have a large gap between the ecological efficiency of the cities and other cities in the province, which indicates that the resources and environmental costs for cities in China to obtain economic benefits are still high. This implies that the economic structure and industrial development need to continue to be optimized and upgraded. At the same time, resource and environmental allocations of the regional differences or inequitable distribution are more prominent. In general, the overall ecological efficiency of Chinese cities is in the medium efficiency range, with a large number of low-efficiency cities and a small number of high-efficiency cities. How to reduce the consumption of resources and environmental pollution, realize sustainable development, and reduce the differences between regions while taking into account economic development is an important topic for the future.

(2) The situation of the four major economic zones. In alignment with China's economic geography and geospatial characteristics, the country is categorized into four distinct economic zones: East, Central, West, and Northeast. As illustrated in Figure 4, the average ecological efficiency values for these zones are 0.638, 0.593, 0.523, and 0.320, respectively, in descending order of East > Central > West > Northeast.

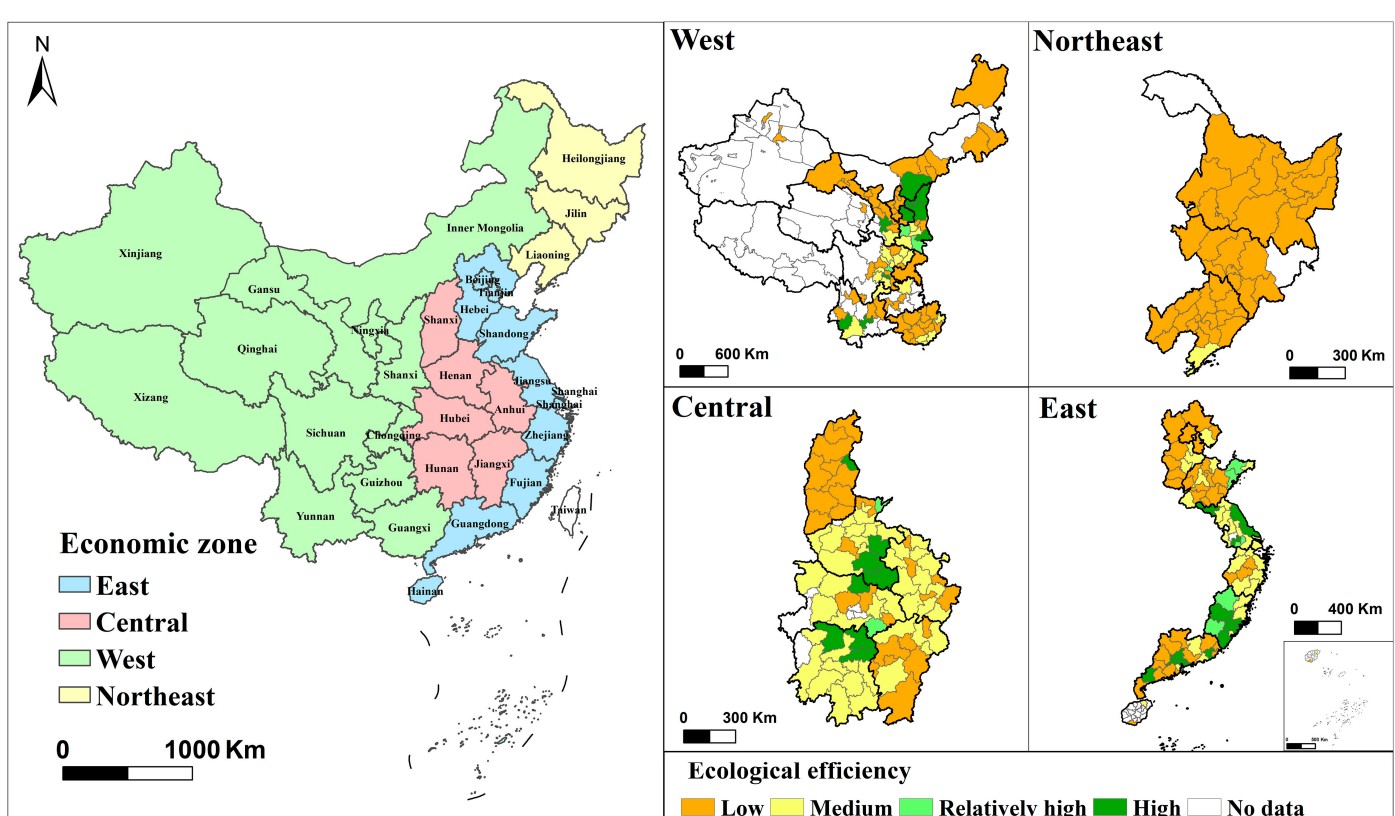

**Figure 4.** China's four major economic subregions and their ecological efficiency.

Eastern Region: This area, as the crux of China's economic and social development, leads with its rapid economic growth and substantial overall output. Boasting abundant labor, capital, technological advances, and human resources, the Eastern region effectively mitigates resource consumption and environmental pollution. Despite having the highest ecological efficiency among the zones, it still operates at a medium efficiency level, indicating room for significant improvement.

Central Region: Rich in resources and labor, the Central region has been absorbing industrial transfers from the East, bolstering its economic growth. However, its economic

foundation and scale pale in comparison to the East. Additionally, it grapples with environmental pollution resulting from these industrial transfers, which explains its lower ecological efficiency relative to the Eastern zone.

Western Region: Geographical, infrastructural, and transportational limitations mark this region. Coupled with a smaller economic scale, later development start, weaker economic base, and a homogenous development model, the Western region's ecological environment is less resilient, resulting in lower ecological efficiency than the East and Central regions.

Northeast Region: Historically a pivotal part of China's economic growth, the Northeast has the lowest ecological efficiency among the four zones. Its focus on heavy industry, alongside a relatively rudimentary development model, has led to significant ecological and environmental challenges. The region's declining development momentum and economic vitality in recent years further exacerbate its position as the least ecologically efficient zone.

(3) The situation of multi-class size cities. According to the Green Paper on Small and Medium-sized Cities [42], Chinese cities are categorized into super-city (more than 10 million), mega-city (5–10 million), large city (1–5 million), and small and medium-sized city (less than 1 million) in accordance with the resident population of the cities.

The results show that the ecological efficiency of cities is positively correlated with city size, exhibiting the hierarchical characteristics of super-cities > mega-cities > large cities > small and medium-sized cities, with their mean ecological efficiency values of 0.675, 0.640, 0.527, and 0.289, respectively, showing an obvious decreasing pattern. Thus, it can be seen that the larger the city scale, the more obvious the agglomeration ability for its resource allocation, and the different resource agglomeration ability plays a role in all aspects of the resource–environment–economy composite system, which ultimately affects the city's ecological efficiency. Therefore, to improve urban ecological efficiency, it is also necessary to consider the differences between cities of different sizes for regulation and management.

*3.2. Spatial Correlation Analysis of Urban Ecological Efficiency*

The overall level of China's urban ecological efficiency is low and shows obvious regional differences. In order to deeply explore the spatial correlation of urban ecological efficiency in China, this study carries out a global autocorrelation analysis based on ArcGIS 10.8 software. The spatial autocorrelation analysis of urban ecological efficiency is carried out by using Formula (2), in which the Moran's I index is 0.225, and the standardized test Z(I) value is 5.736, which passes the test at the significance level of 0.01. The results show that the spatial distribution of ecological efficiency presents significant positive autocorrelation, and the clustering state is obvious.

In order to further identify the typical agglomeration types of urban ecological efficiency, the local spatial autocorrelation analysis of ecological efficiency in cities across the country is conducted using Formula (3), and LISA significance maps are drawn, as shown in Figure 5a.

Cities of the high–high clustering type are mainly located in the south of Henan Province (Figure 5(b1)), the middle of Jiangsu Province (Figure 5(b2)), the southwest of Fujian Province (Figure 5(b3)), the neighboring area of Hunan and Hubei Province (Figure 5(b4)), and sporadically in Shaanxi Province (Yulin) (Figure 5(b5)), Yunnan Province (Puer) (Figure 5(b6)), and so on. The low–low clustering area is distributed in the northern part of China and is relatively concentrated, basically covering the three northeastern provinces (Jilin, Liaoning, and Heilongjiang) and affecting the neighboring cities (Figure 5(c1)).

The phenomenon of low–low clustering also occurs in the northwestern part of Gansu Province (Figure 5(c2)), and in the neighboring areas of Inner Mongolia and Shanxi Province (Figure 5(c3)).

The high–low clustering area and low–high clustering areas are smaller in scope and are mainly distributed in the periphery of the high–high and low–low clustering areas.

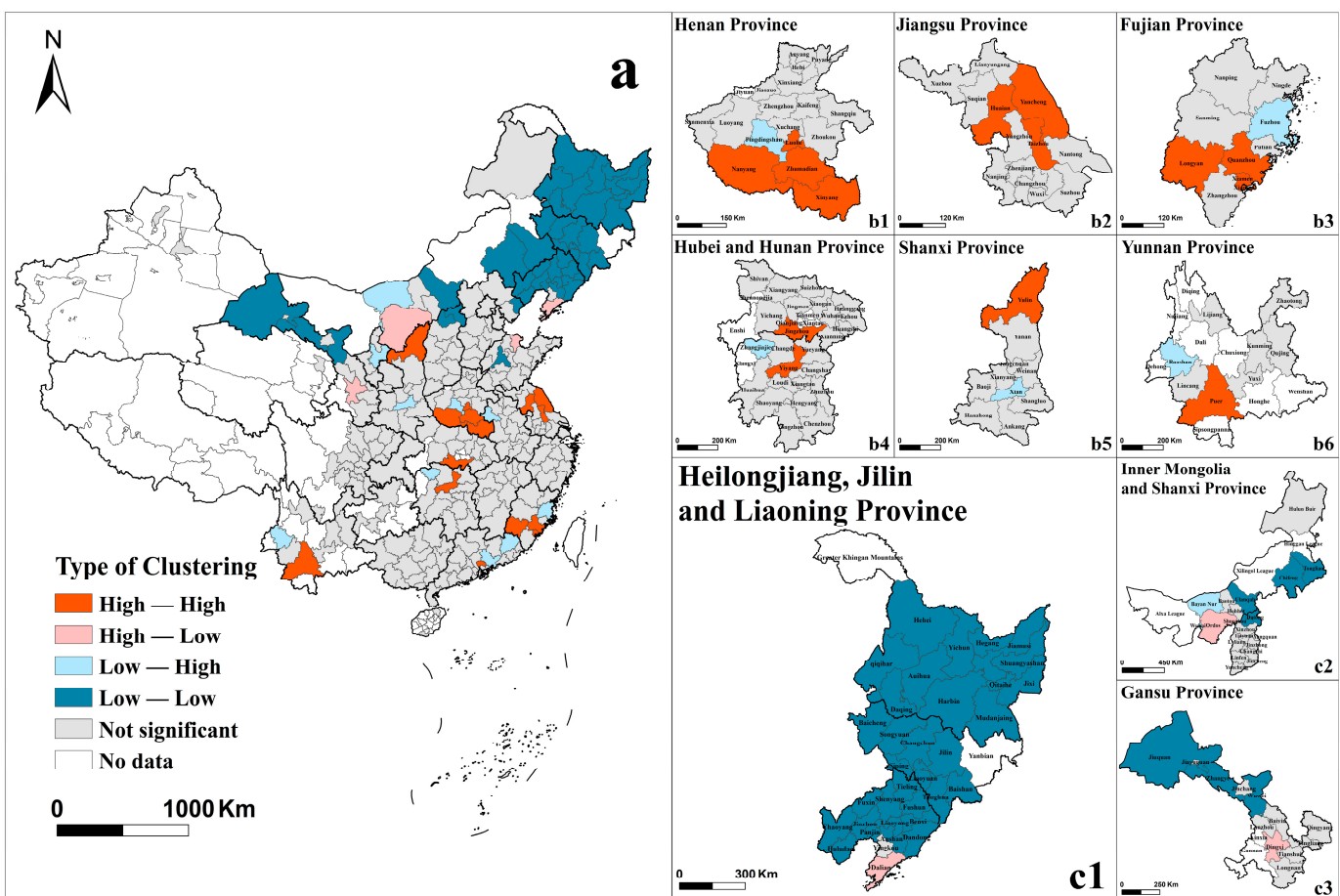

**Figure 5.** Spatial distribution of ecological efficiency LISA clusters in Chinese cities. (**a**). Overall spatial clustering of ecological efficiency in Chinese cities. (**b1**–**b6**). distribution Areas of High–High clustering type. (**c1**–**c3**). distribution Areas of Low–Low clustering type.

From the results of local spatial autocorrelation, it can be seen that both high–high clustering areas and low–low clustering areas have very obvious spatial clustering characteristics. High–high clustering areas are mostly dominated by neighboring cities in the same province. The number of cities in a single cluster is small, which proves that the radiation scope of synergistic promotion of ecological efficiency development is relatively small, and the radiation surface should be expanded in the future to drive more cities in the periphery to improve ecological efficiency. The formation of low–low clustering zones in the three northeastern provinces and surrounding cities shows that the northeastern economic zone has a strong spatial dependence on ecological efficiency. In the future, guided by the idea of "revitalization of northeastern China in the new era", it might still be coordinated to improve its ecological efficiency from the perspective of the three northeastern provinces and the entire northeastern economic zone. Other low–low clustering zones, given the necessity for improving their ecological efficiency, also might be taken into account.

### 3.3. Analysis of Factors Affecting Urban Ecological Efficiency

Before regression analysis, each variable needs to be tested for multicollinearity, as shown in Table 5, the VIF (Variance Inflation Factor) of each variable is less than 5, which indicates that there is no multicollinearity, so regression analysis can be performed directly.

**Table 5.** Multicollinearity test.

| Variable | VIF | 1/VIF |
|---|---|---|
| Economic level | 3.634 | 0.275 |
| Industrial structure | 1.212 | 0.825 |
| Technological level | 2.138 | 0.468 |
| Opening-up level | 1.179 | 0.848 |
| Population density | 1.662 | 0.602 |
| Urbanization level | 2.984 | 0.335 |

ArcGIS software was used to construct the OLS (ordinary least squares) and GWR model. The results showed that the goodness-of-fit (adjusted $R^2$) of the OLS was 0.43 with an AIC value of $-134.87$. In comparison, the GWR model demonstrated a superior goodness-of-fit (adjusted $R^2$) of 0.52 with an AIC value of $-164.84$, indicating better model performance over the OLS. The results of the GWR model were thus selected for analysis. In order to more intuitively portray the local effects of each driver on urban ecological efficiency, the regression coefficients of each indicator in the GWR model results were visualized and analyzed with ArcGIS software. The spatial distributions of the coefficients of each influencing factor were obtained, as shown in Figure 6. The analysis of the regression coefficients' positive and negative values revealed that certain influencing factors exert both positive and negative effects on urban ecological efficiency. The varying proportions of these effects underscore the spatial instability and heterogeneity of these factors. This finding highlights the complex interplay of regional dynamics influencing urban ecological efficiency.

(1) Effect of economic level on ecological efficiency (Figure 6a): The per capita GDP indicator was positively correlated with urban ecological efficiency in China, with coefficients increasing from the southeast coastal areas to the northwest. From the perspective of input and output dynamics, the trajectory of economic development inevitably mandates heightened resource, capital, and factor inputs. However, the beneficial ramifications stemming from enhanced output outweigh the attendant negative repercussions. Primarily, amidst escalating resource scarcity, a gradual transition away from haphazard and unregulated resource and environmental inputs is observed, concomitant with an augmentation in the methodological precision underlying the deployment of diverse factors. Simultaneously, heightened emphasis is placed on the adoption of clean energy sources and the mitigation of environmental pollution. Furthermore, as developmental progression deepens, China's economy of scale effectively enhances the efficiency of resource utilization across various domains. Additionally, the amelioration of economic conditions facilitates increased allocation of capital and human resources towards environmental enhancement initiatives and the adoption of cleaner production methodologies. Consequently, a virtuous cycle ensues wherein elevated economic output is attained through diminished resource and environmental inputs, thus bolstering the foundation for sustainable development endeavors.

(2) Effect of industrial structure on ecological efficiency (Figure 6b): The secondary industry is a vital pillar of the national economy. In this study, we used the proportion of the secondary industry's output value to GDP to measure the industrial structure. The results indicated that ecological efficiency was positively correlated with the industrial structure in over 90% of cities. Only a few cities in northeastern China displayed a negative correlation. Although the secondary industry, including sectors such as manufacturing and construction, consumes a large amount of energy and resources, it has undergone a transformation from high-speed growth to high-quality development driven by national strategies such as sustainable development and innovation-driven growth. This transformation involves a shift from labor-intensive to capital-intensive and knowledge-intensive sectors. The development of high-tech industries has also promoted informatization, which in turn has driven the secondary industry toward higher value-added outputs. The output has shifted from low value-added to high value-added sectors, resulting in significant progress in

industrial upgradation. This transformation is favorable for energy conservation, emissions reduction, and environmental protection. Moreover, although China's economy is gradually shifting toward the dominance of the tertiary industry, the secondary industry is also seeking and improving organic connections with other industries, exploring and promoting the rationalized development of industries. Therefore, a transition toward a more advanced and rationalized secondary industry plays an important role in enhancing urban ecological efficiency. For the few cities in northeastern China where industrial structure and ecological efficiency exhibited a negative correlation, efforts should also be made to accelerate the advancement of industrial structure toward higher quality and rationalization, promoting the development of an enhanced quality-oriented secondary industry.

(3) Effect of technological level on ecological efficiency (Figure 6c): The regression results regarding technological level showed both positive and negative correlations. Positive correlations were mainly observed in regions such as Guizhou, Yunnan, Ningxia, Xinjiang, and northeastern China—areas that were relatively economically underdeveloped. By improving their technological levels, these provinces and cities could achieve significant economic development and effectively enhance ecological efficiency. Conversely, most provinces and cities showed a negative correlation, which might differ from common perceptions. It is often assumed that producers can reduce resource input and environmental pollution through technological advancements. However, in the current Chinese context, the primary goal of improving technological levels remains focused on enhancing economic benefits, with less attention to resource and environmental consumption. Technological progress has not proportionally led to reductions in resource consumption and environmental pollution. Alternatively, one might argue that the cost-effectiveness of improving technological levels is relatively low, as it has not yet achieved a balance between costs and benefits. Moreover, the average proportion of urban technological expenditures to fiscal expenditures in China is less than 2%, which is insufficient to support rapid advancements in urban innovation capabilities and technological levels. It has also not effectively offset the negative effects of long-term extensive resource and environmental consumption. Continuous investment and time are required to improve technological levels in key areas and create economic benefits characterized by lower resource consumption, reduced pollution, and higher quality. Additionally, the transformation of scientific technology into practical productivity is a gradual process and can disrupt the inherent development balance of specific industries and sectors. It exhibits a certain lag and volatility. Therefore, the negative impact of future science and technology on ecological efficiency is likely to gradually decrease and eventually become positive.

(4) Effect of opening-up level on ecological efficiency (Figure 6d): The regression results regarding openness showed both positive and negative correlations, with coefficients increasing from south to north, showing significant regional variations. The southern regions showed a negative correlation, while the northern regions showed a positive correlation. The "Pollution Haven" hypothesis suggests that developed countries transfer highly polluting industries to developing countries, leading to environmental degradation in the receiving countries [43]. Since Southern China opened up to the world earlier and to a greater extent during periods of rapid economic and industrial development, southern and coastal cities attracted a large amount of foreign investment and created numerous employment opportunities, contributing significantly to urban economic growth. As land became scarce and industries upgraded in the core cities, foreign-invested industries moved to surrounding cities, further promoting economic development in these areas. However, as regional economies transitioned toward higher quality, the drawbacks of traditional labor-intensive and pollution-intensive foreign-invested industries became apparent. The negative impacts of extensive industrialization and industrial relocation on resources and the environment surpassed the positive effects on economic growth. This aligned with the "Pollution Haven" hypothesis and was detrimental to improving urban ecological efficiency. Moreover, the negative effects of the "Pollution Haven" hypothesis were persistent and required continuous policy and technological improvements. In contrast, cities in northern

regions were limited by geographical conditions, opened up later, and to a lesser extent, lagging significantly behind their southern counterparts. However, national policies regarding openness also changed. For foreign investment, the focus was not only on meeting economic development needs but also on avoiding excessive resource and environmental consumption. Northern cities, therefore, prioritized attracting foreign enterprises with higher added value, green and clean technologies, and advanced management practices. They reaped the positive benefits of openness on urban ecological efficiency and aimed to reduce or mitigate the negative effects of the "Pollution Haven" through policy measures. With China's changing and implemented policies for openness, advanced foreign investments could be efficiently utilized, while outdated foreign investments continued to move to other countries. Consequently, the impact of openness on urban ecological efficiency was expected to increasingly shift toward positive outcomes in the future.

(5) Effect of population density on ecological efficiency (Figure 6e): The regression results for population density showed a positive correlation, with the regression coefficient generally increasing from west to east. The augmentation of population density denotes the clustering of inhabitants, a phenomenon delineated by Williamson's hypothesis, positing that the impact of population clustering on urban economic growth exhibits a stage-dependent nature. Notably, spatial agglomeration of the population during the initial and intermediate stages of development can profoundly bolster economic efficiency. However, upon surpassing a certain threshold, population agglomeration may attenuate its positive influence on economic growth or even act as a deterrent [44]. Scholarly investigations have underscored the present phase of population agglomeration in China as conducive to economic expansion [45,46]. This sentiment is corroborated by the observable surge in demand consequent to heightened population density, thereby engendering a stimulatory effect on consumption growth. Furthermore, the escalation in industrial concentration and resultant economies of scale are notable outcomes of increased population density. Moreover, the enduring proliferation of knowledge spillovers, fostered by robust competition and collaborative exchange within densely populated locales, furnishes an auspicious milieu for augmenting economic output. Regarding input dynamics, population agglomeration facilitates cost mitigation across various resource inputs, encompassing transportation, transactions, and infrastructure development, thereby curbing superfluous consumption and waste.

(6) Effect of urbanization level on ecological efficiency (Figure 6f): The regression results for urbanization level showed a negative correlation, with coefficients decreasing from south to north. Since the beginning of economic reforms and opening-up policies, China has experienced a remarkable increase in its urbanization level. During the urbanization process, there has been a significant increase in the "quantity" of urban economic benefits. However, this process has largely been characterized by extensive and primary urbanization, which, in many cases, has sacrificed excessive amounts of land, resources, and the environment. There has been relatively less attention to enhancing the "quality" of economic benefits. Consequently, this approach has led to issues such as an irrational land and resource structure, resource allocation constraints, and inadequate urban infrastructure, commonly referred to as "urban diseases." Moreover, with continued population growth and the increasing constraints of the ecological environment, these "urban diseases" have worsened over time. Therefore, this type of urbanization, which prioritizes speed over quality, is detrimental to improving ecological efficiency and achieving sustainable urban development. China's economic development in the new era is transitioning from a high-speed growth phase to a high-quality development phase. To reverse the negative impact of urbanization on ecological efficiency, it is imperative to promote a transformation of urbanization toward higher-quality development. In this process, government and stakeholders should place greater emphasis on the coordination and sustainability of resources and the environment, creating higher economic benefits while simultaneously reducing resource and environmental consumption.

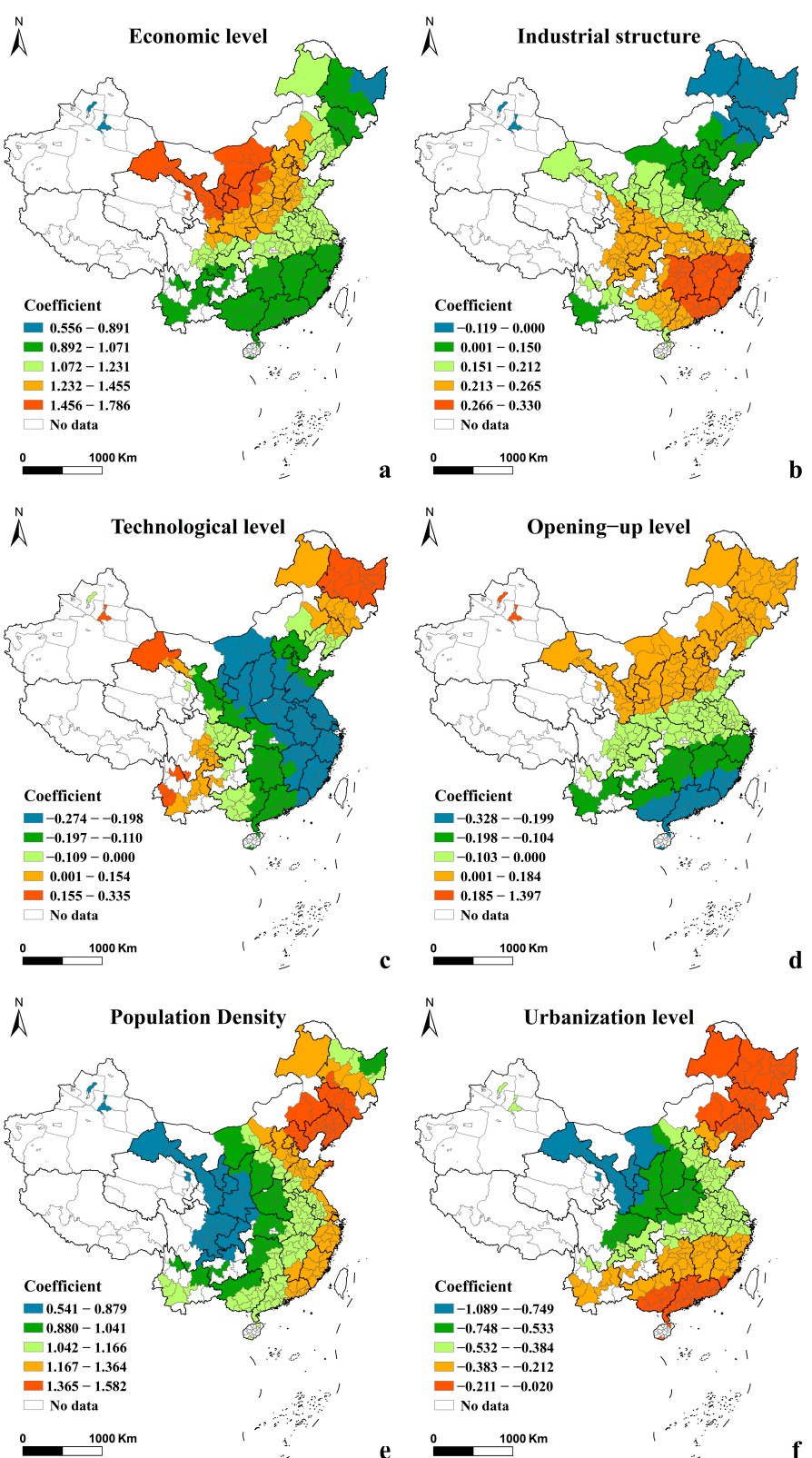

**Figure 6.** Spatial distribution of regression coefficients for impact factors of the GWR model. (**a**). Spatial distribution of regression coefficients for the economic level factor. (**b**). Spatial distribution of regression coefficients for the industrial structure factor. (**c**). Spatial distribution of regression coefficients for the technological level factor. (**d**). Spatial distribution of regression coefficients for the opening-up level factor. (**e**). Spatial distribution of regression coefficients for the population density factor. (**f**). Spatial distribution of regression coefficients for the urbanization level factor.

## 4. Discussion

### 4.1. About the Methodology

To measure urban ecological efficiency, this study adopted the super-efficiency SBM method, which is a widely used mathematical method for efficiency evaluation. Although some scholars have applied other methods, such as SFA or the ratio method, to measure ecological efficiency, these methods have some limitations, such as requiring a small number of study objects, complex data processing, or low discrimination ability. Considering that this study involved 284 cities in China, which is a large and diverse study area, the availability and convenience of data were important factors in choosing the measurement method. Therefore, the super-efficiency SBM method was more suitable for this study.

This study constructed an ecological efficiency evaluation index system using the super-efficiency SBM method, which mainly sets inputs and outputs based on the specific indicators of resource consumption, environmental pollution, and economic outputs. Resource consumption was the input indicator, and economic outputs were the output indicator. As for environmental pollution, different scholars have different treatments. Some scholars considered it as an input indicator, such as Fu and Pan, who calculated the ecological efficiency by treating environmental pollution as an input [32,47]. Some scholars considered it as a "non-desired output" and treated it as an output indicator, such as Ren, who studied the ecological efficiency of urban agglomerations by treating environmental pollution as a non-desired output [27]. This study chose to treat environmental pollution indicators as input indicators for the following reasons. First, from the concept and connotation of ecological efficiency, it can be simply understood as the ratio of economic output to resource and environmental consumption, so this study thought that treating environmental pollution indicators as input indicators was more consistent with the connotation of ecological efficiency. Second, Pan and Long mentioned that pollution was inevitable in socio-economic activities, which was the cost or price that had to be invested in economic output. Therefore, environmental pollution was actually the input of the "cost of environmental degradation", which should be regarded as an input indicator [47,48]. Third, regarding the practical application of the model, it is customary to treat indicators with positive effects as outputs, where larger values are preferable. Conversely, indicators with negative impacts are treated as inputs, with smaller values being more desirable. Based on the above points, this study treated environmental pollution type indicators as inputs.

Moreover, this study used the GWR model to explore the influencing factors of ecological efficiency; it is a new attempt. The GWR model is a local model that can capture the spatial heterogeneity of the study area. Compared with some global models, the GWR model was more reasonable for this study.

### 4.2. About the Results

The results of this study were partly consistent with the studies of Yang and Yan [26,41], who also found that China's overall ecological efficiency was low and showed a spatial pattern of high in the south and low in the north. However, there were some discrepancies in the regional rankings of ecological efficiency. Yan ranked the regions as east > west > northeast > central [41], while this study ranked them as east > central > west > northeast, which was in line with the existing literature that suggested: "the east is the highest, the center is the second, and the west is the worst". The differences could be attributed to various factors, such as the selection of indicators, the measurement methods, and the research periods. The methodology and the evaluation indicator system used in this study were based on the understanding of the connotation of ecological efficiency and the previous research findings, and the data were derived from the official statistics. Therefore, the results of this study had a certain degree of scientific validity and reliability.

This study selected the influencing factors based on the existing research results and used the GWR model to explore their effects on urban ecological efficiency. Factors such as economic development level, population density, and urbanization level were generally

consistent with the previous studies and formed a common view: the improvement of urban economic development level could provide financial support for environmental governance, energy-saving and consumption-reducing technologies, and environmental protection public infrastructure, which could enhance the urban ecological protection capacity and effectiveness, as well as the environmental awareness and green consumption of urban residents [49]. This reflected the importance of coordinating economic development and environmental protection. The increase in population density implied the agglomeration of the urban population, which could improve the resource utilization and sharing rate and reduce pollution. The current rapid urbanization process might bring serious urban problems and should focus on improving the urbanization quality. The effects of factors such as industrial structure, science and technology level, and openness level still need further exploration in future research.

The share of the secondary industry in the gross urban product indicates the industrial structure, while some studies used the share of the tertiary industry. This choice was based on the fact that the secondary industry had been the main driver of China's economic growth for a long time and that it was still an essential part of China's development. The results showed that industrial structure had a positive effect on urban ecological efficiency in most regions of China, except the northeast. This observation contrasts with Zhang's findings, where the secondary industry was identified as having a negative impact on ecological efficiency within five major urban agglomerations [49]. On the other hand, Yan's research presented a different perspective, indicating that the secondary industry actually contributed positively to ecological efficiency [41]. For developed cities, despite the tertiary industry's dominance, the secondary industry remains vital. The retained portion of this sector is primarily composed of high-tech enterprises, which are typically more resource-efficient and environmentally friendly. In contrast, for ordinary cities, the secondary industry was still the main pillar of economic growth. Its contribution to economic output was greater than its consumption of resources and environment. Moreover, under the national policy of industrial upgrading, the pollution caused by the secondary industry has been greatly reduced. The situation is notably different in the Northeast, which was a traditional heavy industry area, it faced the challenges of slow economic development and talent loss. Here, how to revitalize the northeast was also a critical and complex issue in China's development.

For the factor of technological level, most of the existing studies agreed that the technological level had a positive effect on the improvement of ecological efficiency, but there were also some different empirical results. For example, Yang found that the technological level had a significant positive effect on the ecological efficiency of mega-cities, but a negative effect on the large, medium, and small cities [26]. Zhou also showed that technological innovation had a significant inhibitory effect on the green development efficiency of Chinese cities, suggesting that improving environmental quality through technological innovation was currently not feasible in China [28]. Contrarily, the regression results of this study indicated that technological progress positively influenced some cities in the less developed western and northeastern regions, while the rest of the country showed a negative impact. This could be explained by the fact that technological progress had a strong enhancement effect on the less developed cities, particularly in the Northeast, where it serves as a crucial mechanism for optimizing industrial structure and spurring economic growth. However, for most cities, the proportion of science and technology expenditure was relatively low. As a result, the effectiveness of technological progress was limited and its translation into actual productivity is slow. So, this limitation was unfavorable to the short-term improvement of ecological efficiency in these regions.

The opening-up level factor in this study presents a south-negative and north-positive pattern across China, the possible reasons for the formation of this pattern have been discussed above. It is noteworthy that empirical studies have shown different or even diametrically opposite results in the analysis of this factor, although many of them use global models which do not adequately reveal the internal differentiation within the study

area. Consequently, the results of this paper's use of a local model can serve as a reference for an in-depth study.

In addition, different from other countries or regions, China's economic development is very unique and has great differences. In terms of sustainable development, there is a need for correct theoretical guidance, so the question of whether the economic theories of developed Western countries are applicable in China is particularly important. For example, in the study of factors affecting ecological efficiency mentioned above, will the opening-up factor create a "pollution paradise" in China? In the existing research, some scholars in the domain of regional ecological efficiency insisted that there is no "pollution paradise" [28], and some scholars believe that this is a stage of evolution [49]. This paper tries to reveal the impact of opening-up on the ecological efficiency of Chinese cities through a local model at the scale of the whole of China. The empirical analysis shows that the level of opening-up has a positive impact on the cities in the north, while it negatively impacts those in the south. These findings suggest that the "Pollution Paradise Hypothesis" holds true for the cities in the south. This may be caused by the dual influence of geographic location and policy in southern China. However, it is important to note that the scope of this study is limited by its short research period, rendering our findings a preliminary exploration and analysis of the applicability of the hypothesis. Similar theoretical tests such as "Porter's Hypothesis" and the "Environment-Technology Paradox" can be more accurately answered through the analysis of a long time series in future research.

Various studies conducted by scholars outside of China offer valuable insights into ecological efficiency, enriching the scope of our study. For instance, Xia's investigation into Mongolia's ecological efficiency underscores several significant correlations. Economic development, industrial structure, population density, and the adoption of green technology were identified as positively influencing ecological efficiency, while capital investment exhibited a negative correlation [50]. Similarly, Amowine's research spanning 44 economies in Africa revealed a U-shaped relationship between ecological efficiency and economic development. Industrial structure was found to be positively associated with ecological efficiency, whereas total foreign investment and urbanization demonstrated negative correlations [51]. Bianchi's examination of 282 European regions elucidated the positive impact of high urbanization and technological advancements on ecological efficiency [52]. These findings, drawn from diverse geographic contexts, offer valuable reference points for contextualizing and enriching our understanding of ecological efficiency dynamics within China.

*4.3. About the Limitations*

This study, while contributing valuable insights, acknowledges certain limitations that pave the way for future research.

Firstly, the evaluation index system of ecological efficiency needs further improvement and optimization. Apparently, our analysis, in alignment with the existing empirical studies to a large extent, primarily revolves around the triad of resources, environment, and economy. This approach overlooks the social dimension, which is increasingly fundamental considering the complexity of urban human–land systems. To fully examine ecological efficiency, it is necessary to broaden the scope of factors including both ecological and social aspects, therefore offering a more comprehensive insight. In addition, the methodological diversity in measuring ecological efficiency implies a challenge. Currently, there are several methods for measuring ecological efficiency, but there is no unified evaluation method, which is not conducive to comparative analysis and theoretical deepening of ecological efficiency research. Hence, how to comprehensively and scientifically evaluate urban ecological efficiency remains a vital and challenging objective for future research.

Secondly, for the analysis of influencing factors, some results obtained in the empirical analysis of this paper are different from those of previous studies. Due to the complexity of urban development, the intersection and connection between various systems are very close, and this study has not yet analyzed in depth the underlying principles and conduction

paths of the influencing factors. Future research is advised to investigate the potential connections between these factors and dissect the mechanisms and pathways through which they impact ecological efficiency.

## 5. Conclusions

This study explores the ecological efficiency of Chinese cities and its influencing factors through the super-efficiency SBM model and the GWR model, and finally draws the following conclusions. First, China's overall low ecological efficiency in 2019 indicates that China is currently in a period of adaptation and transition from rapid economic growth to high-quality development. The crude development model characterized by high consumption and pollution formed in the previous stage has not yet been completely changed. Secondly, there are obvious disparities in the sustainable development levels across different economic regions and among cities of varying sizes. Third, urban ecological efficiency is spatially relevant, with the direction and intensity of influencing factors displaying spatial heterogeneity.

The following insights can be gleaned from the aforementioned findings: (1) Enhancing Resource Utilization Efficiency. A multifaceted approach is necessary to improve resource utilization efficiency. This involves modernizing production technologies in key industries, scrutinizing new projects to minimize resource consumption and pollution, and promoting sustainable urban development through policy enactment and public awareness campaigns. (2) Prioritizing Science and Technological Innovation. Science and technological innovation are paramount for sustainable development. Increasing investment in research and development, particularly in green technology, is crucial. Additionally, fostering innovative talent through targeted training programs and creating an enabling environment for innovation are essential. (3) Enhancing Urbanization Quality. Improving urbanization quality requires tailored approaches for each city. This includes enhancing the scientific rigor of urban planning, optimizing urban spatial configurations, and coordinating population dynamics, economic activities, and environmental resources for sustainable urban development. (4) Reforming Ecological Construction Paradigms. Recognizing urban agglomerations as crucial for China's development underscores the need to integrate ecological construction and environmental preservation efforts. This involves addressing management inefficiencies between cities, establishing regional resource and environmental co-construction mechanisms, and leveraging urban agglomerations to promote regional sustainability while considering regional heterogeneity and adhering to categorized guidance for effective implementation.

**Author Contributions:** Conceptualization, C.X.; Methodology, Y.L.; Data curation, J.P.; Writing—original draft, J.P., C.X. and D.C.; Writing—review & editing, C.X.; Supervision, Y.L.; Funding acquisition, Y.L. All authors have read and agreed to the published version of the manuscript.

**Funding:** This research was funded by National Natural Science Foundation of China grant number: 41771096. And the APC was funded by it.

**Institutional Review Board Statement:** Not applicable.

**Informed Consent Statement:** Not applicable.

**Data Availability Statement:** Data is contained within the article.

**Acknowledgments:** Many thanks to Xiankun Yang for his guidance on English academic writing.

**Conflicts of Interest:** The authors declare no conflict of interest.

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
