# Peer review of "Unveiling the Patterns and Drivers of Ecological Efficiency in Chinese Cities: A Comprehensive Study Using Super-Efficiency Slacks-Based Measure and Geographically Weighted Regression Approaches"

_sustainability, doi:10.3390/su16083112_

Round 1

Reviewer 1 Report

Comments and Suggestions for Authors

The approached subject is a very interesting one, taking in fact the phenomena of urban sprawling and the correlated increase in population. Indeed, the question of ecological efficiency in urban areas, from the largest but going at least to those of medium dimension is very important.

I personally do not agree with the fact that economic development and especially population density can positively influence ecological efficiency, they are both factors leading to pollution, agglomeration and other.

Indeed, proper measures should be taken based on such studies, to reduce the urban pressure.

 The definition from 60-64 rows seems to not consider borders. Let us remind the idea of the butterfly effect.

If I would consider ecological efficiency, I would go back to the hunter-gatherer period.

I appreciate the taking into account of various methods, but also need to cite the authors – studies have yielded diverse outcomes for a given study area in a specific period.

106-107 – Agree, sometimes we need to extend our focus. Maybe also comparing urban expansion to rural areas.

Passing over the statistical methods and overall model used, how was the sewage discharge determined?

I would prefer to not mention political aspects in the paper (ex. Taiwan).

Row 274-275 – for the sake of scientifically and accuracy, those data from  pandemic years should be included in the analysis.

279-280 – How have the outliers have been adjusted. Based on what?

Results – a comparison with some American, European, Asian, African cities would do some good for the paper.

There seems to be put accent on the economical benefit vs ecology. Food for thought. Seems like more resource use, more pollution, more production, can be interpreted as ecology efficiency.

The relation between human density, pollution, and sustainability is a tricky one. Sometimes I would go back to classical methods instead of using statistical methods. Data, real data.

Reviewer 2 Report

Comments and Suggestions for Authors

The authors use the concept of urban ecological efficiency and an index for ecological efficiency evaluation to assess and compare the level of sustainable across different cities. Regression model is also developed to better understand the impact of various factors on ecological efficiency, revealing some interesting spatial patterns. It is an interesting study with potential policy implications but this study should not be published in its present form, in particular in terms of not sufficiently describing the research methods. My comments to each part of the manuscript are below.

The abstract reflects the focus of the study and briefly summarises the main findings.

In the introduction the information about Chinese national policies seems partly unnecessary. It would be more interesting to focus on regional and local policies. The theoretical background is well presented, with relevant references. The novelty of the approach and its impact on the development of the ecological efficiency concept should be better outlined.

The methods are not clearly presented. To begin with, they are not sufficiently discussed in reference to previous studies, also to show what methodological advancements this study brings. This should be further developed. The description of the construction of the ecological efficiency measurement indicator system is very general. Please provide more details. The description of selecting the factors affecting ecological efficiency is also very vague (for example, what does it specifically mean ‘considered data availability to investigate these specific external influences’?). It is not clear which parameters from previous studies were used or which were excluded simply due to lack of data (in such a case it should be discussed how it affects the analysis). It is hard to asses this process without a solid description.

It is hard to interpret the results without more detailed description of the methods. The description of the results should be better organised.

Policy implications could be more specific. The applicability of the research approach in practice to support developing more sustainable policies can be further explored.

Further minor remarks:

Please explain the abbreviation SBM in the introduction

-       -  There are some formatting issues (e.g. the tables) and some passages could be divided into shorter paragraphs for more convenient reading

-       -  A professional proofreading of the manuscript could be helpful

Comments on the Quality of English Language

-        A professional proofreading of the manuscript could be helpful.
